# Effect of Bentonite Addition to Pedro Ximénez White Grape Musts before Their Fermentation with Selected Yeasts on the Major Volatile Compounds and Polyols of Wines and Tentative Relationships with the Sensorial Evaluation

**DOI:** 10.3390/molecules27228057

**Published:** 2022-11-20

**Authors:** Raquel Muñoz-Castells, Jaime Moreno-García, Teresa García-Martínez, Juan Carlos Mauricio, Juan Moreno

**Affiliations:** Department of Agricultural Chemistry, Edaphology and Microbiology, Marie Curie (C3) and Severo Ochoa (C6) Buildings, Agrifood Campus of International Excellence CeiA3, University of Córdoba, 14014 Córdoba, Spain

**Keywords:** wine, bentonite, yeast, fermentation, volatile compounds, statistical analysis

## Abstract

In this work, we study the effect of bentonite addition to the grape must before alcoholic fermentation on the chemical composition and sensorial profile of the obtained wines. Fermentations were carried out with two *Saccharomyces cerevisiae* commercial active dry yeasts treated or not with bentonite and were compared with a control wine obtained by spontaneous fermentation (using the grape must microbiota). Several significant effects on the chemical and sensorial attributes were established by statistical treatments. The selection by multiple variable analysis of seven volatile molecules (ethyl acetate; methanol; 1-propanol; isobutanol; 2-methyl-1-butanol; 3-metyl-1-butanol and 2-phenylethanol) provided several footprints that provide an easy visualization of bentonite effects on wine volatile compounds. A Principal Component Analysis carried out with all the compounds quantified by Gas-Chromatography revealed that the first two Principal Components explain 60.15 and 25.91%, respectively, of the total variance and established five groups that match with the five wines analyzed. Lastly, predictive models at *p* ≤ 0.05 level for the attributes sight, smell and taste were obtained by Partial Least Squared regression analysis of selected chemical variables.

## 1. Introduction

Chemical and sensory characterization are two general aspects when studying quality differences between wines, and can be established by chemical analysis and organoleptic evaluation, respectively. The analysis of some classical variables and the quantification by instrumental analysis of some important secondary metabolites are used for the classification of wines or to evaluate the effects of different oenological treatments in an objective way [1]. There are many factors to consider when evaluating the sensorial quality of wines, but nowadays, those related to sight, smell, and taste are the most important. Thus, the visual aspect is the first sensation causing consumers to reject the product. Currently, the occurrence of hazes and deposits in bottled wines that affect their clarity and transparency is a major concern in the wine industry, and is responsible for large monetary losses every year [2].

The appearance of turbidity in white wines is commonly due to the presence of tartaric acid salts and/or unstable proteins. Potassium acid tartrate and calcium tartrate are the two scarcely soluble salts that precipitate in wine at low temperatures. This break in clarity is usually avoided by subjecting the finished wine to a low-temperature treatment for a specific period. However, the unstable proteins causing haze and sediments due to the processes of denaturation, aggregation, and flocculation are commonly removed by using bentonite as fining agent on the finished wines, after the fermentation process and prior to their bottling [3,4,5]. No adequate evaluation of this practice on wine chemical quality has been carried out [6].

Bentonite is the commercial name of a clay material that is commonly used in winemaking [7]. Novel fining agents such as seaweed polysaccharides, chitin, zirconium dioxide, and packed-bed cation exchangers, as well as ultrafiltration techniques, have been explored in recent years. However, these agents must meet several criteria such as being cost-effective, nontoxic, and innocuous to wine quality [8,9]. Bentonite has adsorption and cation-exchange properties and forms colloidal dispersions in water or hydro-alcoholic solutions with a negative electrostatic charge. These particles attract other positively charged colloids such as proteins (at wine pH), resulting in a mutual flocculation and the subsequent precipitation to the bottom of the container [3,9]. Nevertheless, use of bentonite as a fining agent shows some unfavorable effects on wine composition. Studies carried out by Catarino et al. (2008) [7], Lira et al. (2014) [10] and He et al. (2020) [4] evidence a significant influence on proteins, amino acids, biogenic amines, and polyphenols contents. In addition, fining agents decrease the content of some wine volatile compounds such as ethyl esters, acetates, and higher alcohols, thus affecting its aroma [11]. In this sense, Pocock et al. (2011) [12], Lira et al. (2015) [11] and Horvat et al. (2019) [13] recommend the addition of bentonite during alcoholic fermentation to prevent protein haze because a smaller amount of product is required and consequently the removal of aroma compounds is apparently lower. Otherwise, Ayestaran et al. (1995) [14], Puig-Deu et al. (1999) [15] and Lambri et al. (2012) [16], show that the bentonite addition to white grape musts and its subsequent separation have a favourable impact on protein stabilization and preservation of varietal aromas and the overall quality of wines. In contrast, Somers and Ziemelis (1973) [17] and Puig-Deu et al. (1999) [15] established that bentonite treatment of finished wine is more effective in stabilizing it from the protein hazard. Lastly, according to Vela et al. (2017) [5], clarification with bentonite to obtain wines with protein stability can be carried out at almost all stages of the winemaking process. Regarding this, Moreno and Peinado (2012) [9] recommend the addition of bentonite to previously decanted must from white grapes, and its remaining during fermentation until completion, to obtain the following advantages: (1) a more compact lees production, and consequently, less liquid lost; (2) partial elimination of tyrosinase, which leads to certain protective actions against oxidation; (3) stimulation of fermentation by providing a support for the yeast; (4) adsorption of traces of fungicides transferred from the vineyard; (5) contribution to the production of finer, clearer wines, without interfering with aroma, and (6) reduction of the number of operations required during the winemaking process. Therefore, its dose and the addition time during winemaking, play an important role in determining the wine sensorial properties [13] and further, on the elimination of wine off-flavours [18].

On the other hand, it is well known that the wine aroma is strongly influenced by the specific metabolism of the strain of *Saccharomyces cerevisiae* carrying out the fermentation process [19,20]. Therefore, the selection of different strains capable of enriching the aroma complexity of wines, and thus their quality, is expanding currently in the wine industry [21].

The use of selected active dry yeasts (ADY) is the current trend among winemakers working in new wine-growing locations and elaborating new wine types. However, the use of indigenous wild yeasts (WY) is preferable to elaborate the traditional wines in wine-growing areas of recognized quality. In any case, the contribution of indigenous non-*Saccharomyces* yeasts to organoleptic differentiation and wine quality is more significant in the first stage of alcoholic fermentation when the ethanol content is still low. It is also known that the yeast species with high ethanol tolerance predominate over the sensitive ones when the ethanol content reaches around 4–8% (*v*/*v*), and are able to finish the alcoholic fermentation, consuming the fermentable sugars and producing dry wines. For this reason, winemakers supplement non-treated musts (containing wild yeasts) with pure cultures of selected ADY, allowing them to develop new wines with diverse aromatic profiles from the same grape variety [22]. The use of this technology also facilitates an adequate control of the elaboration process and allows the diversification of products to adapt wine production to the current consumer demands [23]. As different strains have different fermentation kinetics [20], it is important to study the effect of bentonite on them and there are few studies that use statistical analyses to establish these relationships. In this regard, the instrumental analysis of secondary metabolite contents in wines entails a large amount of variables and data, so it is appropriate to use a multivariate analysis approach. Through these statistical analyses, it is possible to extract relevant, objective, and useful information through experiments carried out on complex matrices such as wine. In addition, when studying the correlations between variables, more relationships can be established, according to the purposes of the research [24].

Lastly, it is well known that the wine industry is subjected to a high degree of competitiveness. The advances in new technologies and the knowledge developed in the last decades make it unavoidable to look for innovations which can increase productivity while decreasing costs without reducing wine quality. In this sense, the proposed research contributes to these advances by studying the effects on the analytical and sensory quality of wine when bentonite is added to the must and remains during the fermentation. The results obtained can help winemakers to make their decisions, improving the incorporation of technological advances and new production processes into the wine sector, which show a low innovation level [25].

The discrepancies described above regarding the appropriate moment for the addition of bentonite, together with the use of active dry yeasts, open new challenges concerning the effect of this fining agent on the fermentation process and its interaction with aroma compounds produced by yeasts.

This study aims to (1) chemically characterize and sensorily evaluate the wines obtained by adding bentonite and different yeast starter cultures to the same grape must before their alcoholic fermentation, (2) study the effect of the interaction of yeast-bentonite on the major volatile compounds and polyols, and (3) relate the chemical compositions to sight, taste and smell attributes by multivariate statistical analysis.

## 2. Results and Discussion

### 2.1. Winemaking Variables

The fermentation process was followed by measuring the density, which decreased from 1088 g L^−1^ to 987 g L^−1^ after 15 days. Although the rate of fermentation carried out with the wild yeasts (WY) was always slower, under all the conditions tested the fermentation was finished in 15 days, when the density value was close to 990 g L^−1^. These results show a low influence of bentonite on the fermentation kinetics (Figure 1).

Table 1 shows the values for the variables used for wine characterization and the results obtained from the Student–Newman–Keuls test at *p* ≤ 0.05 significance level. Only total acidity and reducing sugars show 5 homogeneous groups (HGs), in a clear correspondence with the five wines, while the remaining variables show 4 HGs with the exception of ethanol content with only 3 HGs. The lowest value of reducing sugars was obtained from the fermentation carried out with the PDM and Caracter active dry yeasts, both without bentonite addition (Y1B− and Y2B− respectively). The two wines obtained using Y2 yeast have total acidity values higher than the remainder. The addition of bentonite increases acidity values in all wines.

Y2 wines show the highest ethanol content, reaching 12.5% (*v*/*v*) versus 12% in the remaining wines. It is observed that the addition of bentonite decreases the ethanol content by about 0.2% (*v*/*v*). The pH values are significantly different in wines obtained with the two commercial yeasts (3.22 for Y2 and 3.43–3.49 for Y1) and the WY wine (3.46). According to Xifang et al. (2007) [26], high ethanol content eases the engagement of proteins in the bentonite structure, by widening the channel between its middle layers, thus increasing the protein adsorption. Low pH values modify the ionization degree of proteins and also the bentonite surface electric charge, influencing its cation exchange capacity and consequently the adsorption power of cations and proteins [3].

Volatile acidity and total acidity values discriminate between the two commercial yeasts, but only wine from Y1 is affected by the addition of bentonite. The volatile acidity reaches its highest value in Y1 wines, being twice as high as in Y2. The titratable acidity value is highest in Y2 wines.

One of the parameters that is clearly affected by the addition of bentonite is the reducing sugars content. Both wines obtained by ADYs show low content when bentonite is added, revealing a more effective fermentation process. This effect is probably due to a detoxification of the fermenting medium, as well as their action as a physical support for the formation of yeast colonies. Similar results are reported in a recent review [27], when bentonite is added as a clarifying agent for stabilization of sweet ice wines.

The absorbances measured at 420, 520 and 620 nm are related to yellowish-brown, reddish-brown and bluish shades, respectively. All of them show 4 HGs and the effect of bentonite is different according to the yeast used (Table 1). Color intensity (CI) is given by the sum of the absorbances measured at 420, 520 and 620 nm, and Total Polyphenol Index (TPI) is a measure of absorbance at 280 nm in samples diluted 1:50, after multiplying by this dilution factor. Values of CI range from 0.222 to 0.333 (Table 1) and are considered as normal for white wines. The values for TPI are also considered common for this type of wine, for which the usual values range from 4 to 10 (Table 1). However, significant differences were found for this index between the two yeasts and, in turn, between Y1 with and without bentonite. TPI changes between yeast strains tested may be due to the interactions of phenolic compounds with cell wall proteins, the co-flocculation processes and/or the specific metabolic activity of each strain [28]. Therefore, the addition of bentonite could have as a secondary effect a decrease in the content of phenolic compounds with some yeasts. These losses are associated with losses in astringency, bitterness, and hotness, and consequently affect in a positive way the texture and mouth feel of the resulting wine [4].

### 2.2. Effects on the Major Volatile Compounds and Polyols

This group of secondary metabolites come from alcoholic fermentation by yeasts and can be classified into highly volatile molecules, such as alcohols, carbonylic compounds and ethyl esters with a low molecular weight, and short-volatile molecules, such as polyols.

Among the six alcohols quantified, the isoamyl alcohols (3-methyl-1-butanol and 2-methyl-1-butanol) and isobutanol show 5 HGs (Table 2), in a clear dependence on the variables of bentonite addition and yeast tested. The remaining alcohols—methanol, 1-propanol and 2-phenylethanol—show 4 HGs and have higher levels in wines treated with bentonite as pre-fermentative agent and Y2 (Caracter). In this context 2-phenylethanol is the only alcohol with a pleasant rose odor descriptor and the WY (Wild Yeast) wine shows the highest content, followed by Y2 wine with bentonite addition.

As between the carbonylic compounds, acetaldehyde contents show also 4 HGs and the wines obtained with bentonite addition have lower contents, which is favorable to the sensorial quality. However, acetoin, an acetaldehyde derived compound, has 4 HGs, but its content decreases with bentonite in Y1 (PDM) and increases in Y2, showing a bentonite-yeast interaction.

The polyols 2,3-butanediol in both their forms (*levo* and *meso*) only show 2 HG each and wines from Y1 fermentation have significantly higher contents than Y2 and WY, although there is no significant differences due to the addition of bentonite in each of them. Glycerol shows 4 HGs, and their content is highest in Y2 wines and increases with the addition of bentonite in all cases. There are many short volatile compounds whose contents are strongly affected by the yeast strain that enable differentiation of the resulting wines [20].

Among the quantified ethyl esters, ethyl acetate shows 4 HGs and a low content in all wines, with Y1 having the lowest content and decreases with the addition of bentonite. Diethyl succinate does not show significant differences between any of the wines (Table 2), mainly because this compound increases slowly during wine ageing, and is a good index of this process [2].

These results indicate that, although the contents of these compounds are yeast dependent, they can also distinguish between the effect of bentonite addition for each yeast tested.

Methanol and the two isoamyl alcohols show higher concentrations in both yeasts when the wine was made in the presence of bentonite. Higher alcohols have a positive effect on the quality of wines when their contents are below 400 mg L^−1^; this is related to their esters with acetic acid, which are important to the wine aroma, giving a fruity odor note [29].

There are some compounds analyzed that only show differences in the presence of bentonite for a single yeast, as with ethyl acetate and 1-propanol in Y1. The same pattern is reported in Y2 wine and the acetaldehyde and 2-phenylethanol contents, showing a different behavior towards bentonite addition for each yeast tested.

There are few studies about the effects on the wine volatilome of bentonite addition to the grape-must before the fermentation process. Nevertheless, the addition of bentonite after the alcoholic fermentation, as a fining agent, is a well-established winemaking practice, causing the removal of some aroma compounds from the wines. This effect is explained by the adsorption of aroma compounds on the surface of colloidal particles formed by bentonite linkage to proteins, followed by its mutual flocculation, because only a few odor active molecules are removed by direct adsorption over bentonite [3,30,31].

As suggested by Ubeda et al. (2021) [2], some compounds can be directly adsorbed by the grape proteins which are removed from the wine through mutual flocculation with bentonite. In contrast, other secondary metabolites produced by fermenting yeast will have more affinity to some other soluble yeast proteins but are not flocculated to the bottom of the vessels, establishing differences between yeasts. Furthermore, in the yeast autolysis after the fermentation and sedimentation processes, proteins, lipids, nucleic acids, and polysaccharide compounds are released to the wine [29], significantly changing the contents of major volatile compounds. These results can be explained by the combined effects among volatile compounds-yeast proteins-bentonite adsorption, taking into account their electrostatic charge and interactions [32].

### 2.3. Statistical Analysis of the Data Matrix: Footprints and Principal Components Analysis

Chemical footprints (sunray plots) for each wine extracted from the MVA of compounds that show 4 and 5 HGs (ethyl acetate, methanol, 1-propanol, isobutanol, 2-methyl-1-butanol, 3-methyl-1-butanol, and 2-phenylethanol) are plotted in Figure 2. These plots allow quick, easy, and useful visualization of the differences between the wines obtained with different starter cultures and with or without the addition of bentonite. Wines made with the same yeast show similar footprints, but there are differences with or without the addition of bentonite, and in comparison with wine from wild yeast. This last wine, used as control to compare the yeast and bentonite effects on the quantified volatile compounds, shows higher contents in 2-phenylethanol, the only higher alcohol with pleasant odor aligned to rose and honey. Y2 (Caracter) wines show higher contents in ethyl acetate, 2 and 3-methyl-1-butanol and 2-phenylethanol than Y1 (PDM) wines, increasing their contents by the addition of bentonite. This agent also increases the contents in all 7 compounds selected for Y2 wine sunray plots.

Figure 3 shows the PCA results for the 13 volatile compounds and polyols quantified in all wine samples. Two components with eigen values above 1, accounted for 86.06% of the total variance (PC1 60.15% and PC2 25.91%). The 2,3-butanediol (*levo* and *meso* isomers), ethyl acetate, acetaldehyde, 3-methyl-1-butanol and isobutanol are the 6 most important contributors to PC1 with coefficients higher than 0.3. In addition, glycerol, acetoin, 2-phenylethanol and the 2-methyl-1-butanol contribute to the PC2 with coefficient values between 0.5 and 0.3. These PCs establish five groups, according to the scores calculated for the five wine types. PC1 scores group the wines according to the yeasts used for alcoholic fermentation, with Y1 wines having wines the highest and most positive scores, followed by WY and Y2, both with negative values. PC2 groups wines according to the use of bentonite, showing the wines from Y2 having a smaller separation between them than those from Y1 with or without bentonite addition. Wines from Y1 and with bentonite have lower scores in PC2 than those without its addition. The PCA was able to distinguish each one of the wines according to their scores in the two first PCs calculated using the absolute contents of volatiles and polyols. The yeasts factor contributes to the variance, as previously observed by other authors [33,34,35]. Furthermore, for each strain it is possible to differentiate the wine with bentonite from the wine without bentonite addition by their content in the major volatile compounds and polyols.

Several studies show that the effect of bentonite addition to white grape must is variable and dependent on grape variety. Regarding this, Armada and Falque (2007) [30] found no significant differences but Lambri et al. (2010) [3] reported clear decreases in ethyl butyrate or hexanoate associated with bentonite. Even higher quantities of some compounds in musts fined with bentonites are found in other studies [10,11,36]. Vela et al. (2017) [5] found in Sauvignon blanc that only butyl acetate, ethyl hydrocinnamate and ethyl cinnamate showed significant differences linked to bentonite treatment, while other ethyl esters and acetates did not show significant differences. These results agree with those reported previously [3,16], suggesting that the vintage and bentonite type are major sources of variability.

In addition, the results obtained in the present work show that bentonite has different effects according to the inoculum format used for the fermentation of Pedro Ximénez grape musts, providing different footprints at a significant level (*p*-value = 0.05) for the content in methanol, 5 higher alcohols and ethyl acetate in wines.

Furthermore, the use of advanced statistical tools such as Principal Component Analysis, applied to all volatile compounds and quantified polyols, enables differentiation between wines treated with and without bentonite for each inoculum tested, based on the scores of each wine. Through this analysis, a smaller difference is obtained between the wines from PDM yeast than between wines from Caracter yeast with or without bentonite.

### 2.4. Sensorial Analysis of Wines

Table 3 shows the means, standard deviations, HGs and ANOVA results calculated for the sensorial attributes (sight, smell, taste and overall quality) evaluated. These data indicate that there are not significant differences for taste between wines obtained with the 3 yeasts tested. Nevertheless, the overall quality shows differences at *p* ≤ 0.05 level, while sight, smell and total points show differences at *p* ≤ 0.01 in relation to the yeasts. The Kruskall-Wallis tests carried out to study the effect of bentonite addition on the scores of wines from two ADYs tested do not show significant differences. The MANOVA (at *p* ≤ 0.05 level) established three HGs for sight and smell in accordance with yeasts tested, two HGs for overall quality and only one HG for taste.

All wines were scored by the sum of attributes evaluated for sight, smell and taste (overall quality) as good wines (70–79 points) as set out in the OIV scale except for the wine made with the wild yeast with 66 points. The highest score was reached by Y1 (PDM) without bentonite (77 points) followed by the wine made by Y1 with bentonite (76 points). The two wines made with Y2 (Caracter) (with or without bentonite) were scored with 73 points.

The differences observed in smell scores of tested wines may be due to the effect of bentonite on the contents of major volatile compounds excreted to the medium. In this respect, it is known that high amounts of isoamyl alcohols may block the perception of fruit attributes [2]. Some authors suggest that grape proteins have little or no effect on the depletion of grape-derived aroma after bentonite clarification. However, yeast-derived mannoproteins have a certain protective effect on aroma compounds of bentonite-treated wines [37]. On the other hand, the negative effect of bentonite on foam stability and persistence of sparkling wines has been established, mainly due to the elimination of grape proteins, while yeast-derived proteins favor these aspects [29]. In addition, their interactions with aroma compounds confer sweet and floral flavors [6].

In terms of sight or wine appearance, the panel tasters did not notice significant differences between the wines treated with bentonite from those without its addition. The presence of bentonite during the fermentation process helps to clarify the still wines, making them less turbid and therefore with a better appearance. Bentonite acts by compacting the volume of yeast lees deposited at the bottom of the container, thereby diminishing the volume of wine to filter and contributing to the sustainability of the wine industry.

### 2.5. Tentative Correlations between Sensory Attributes and Chemical Variables

Chemical and sensorial data matrix were subjected to a MVA to calculate simple regression coefficients, in order to establish tentative relationships among chemical variables and scores given by sensorial tasters for taste, sight, and smell properties of wines. The obtained correlation coefficient matrix establishes significant relationships at *p*-value ≤ 0.05 of taste scores with acetic acid (0.0239), reducing sugars (0.0004), methanol (0.0293), 1-propanol (0.0013), 2,3-butanediol (*levo*) (0.0173) and 2-phenyethanol (0.0053). This matrix also shows relations at this significance level between sight and acetic acid (0.0368), reducing sugars (0.0166), absorbance at 280 nm (0.0417), 1-propanol (0.0024), 2,3-butanediol (*levo*) (0.025) and 2-phenyethanol (0.0041). Lastly, significant correlation coefficients are obtained between smell and diethyl succinate (0.0302) and glycerol (0.042). Nevertheless, these statistical results do not agree with the well-known impact that some of the quantified variables have on the sensory attributes of wines [38]. As an alternative, a PLS analysis was carried out to establish a predictive model based on the data of each sensorial attribute and specific chemical variables selected by their impact on the taste, sight and smell senses. In this way, the absolute contents of chemical compounds or fractions quantified were considered as independent variables (X- data matrix) and the sensorial attributes as dependent variables (Y- data matrix). The PLS analysis combines the features of PCA and multiple regression analysis, first extracting a set of components (or latent variables, LV) that explains as much as possible of the covariance between the dependent and independent variables. Afterwards, a regression step predicts the values of the dependent variables by combining the independent variables. Therefore, the PLS analysis captures the variance and obtains the correlation [39]. The selection of the number of LVs to obtain a good PLS predictive model was carried out by Cross Validation (CV) analysis, focused on improving the prediction goodness (Q2) and the percentage of explained variance of the sensorial attribute predicted, minimizing the Root Mean Square Error (RMSE) of the CV. The PLS analysis gives statistically significant predictive models for the dependent variables: taste, smell, and sight, as shown Table 4. Based on the average Prediction R-Square, the best model for taste is the one that uses 3 LVs based on 4 chemical variables, explaining 99.89% of total variance and with 0.2 as the RMSE of prediction. The PLS model for sight was able to explain 65.45% with 2 LVs based on 6 variables and a RMSE value of 0.133. Lastly, the PLS model for smell explained 72.68% of the original variance with 2 LVs obtained from 10 chemical variables and 0.133 as the RMSE value.

## 3. Materials and Methods

### 3.1. Wines and Winemaking Conditions

The must was obtained from the Pedro Ximénez grape variety growing in the Montilla-Moriles winemaking region (Córdoba, South Spain) and harvested with 210.1 g L^−1^ of sugars content and a pH value of 3.8. The must was subjected to a pre-fermentative treatment by adding 1.5 g L^−1^ of tartaric acid and dipotassium meta-bisulphite (K_2_S_2_O_5_) to reach a final concentration of 50 mg L^−1^ in sulphur dioxide (SO_2_) content. The effect of bentonite (5 g hL^−1^) added to the musts before their fermentation with Pasteur Prise de Mousse (PDM) or Caracter, active dry yeasts (ADY), which were used as starter cultures, were compared with the control wines obtained without bentonite addition and with the wine obtained by spontaneous fermentation with wild yeasts. This fermentation was used as negative control to compare the effects of both variables on volatile compounds, polyols, and sensory analysis. This type of control is widely used by winemakers in relevant studies to verify the effect of the tested variable [5,40]. PDM and Caracter are two commercial *S. cerevisiae* ADYs provided by Agrovin™. The first was isolated from the Champagne region (France) while Caracter was from Rioja (Spain). In this study, wild yeast was called as “WY”, PDM as “Y1” and Caracter as “Y2”, including the symbol positive (+) when bentonite was added and negative (−) in the absence of bentonite.

Bentonite suspension was prepared by adding hot water (37 °C) to bentonite (ratio 10 to 1). The mixture was stirred for 30 min and left at room temperature for 24 h. Then, it was added to the musts (5 g/hL) according to the recommendations of experienced winemakers, who use bentonite in Pedro Ximenez healthy grape must fermentations. This dose can be increased for rotten grapes or decreased in the case of the second fermentation of sparkling wines. Our work has added bentonite to the must before fermentation and left it in the fermenting must until the spontaneous stabilization of wine at low temperature.

Each ADY used as starter inoculum was rehydrated in accordance with the provider recommendations. After this, each one was conditioned in a medium containing 50 g L^−1^ glucose, 2.8 g L^−1^ tartaric acid, 2.4 g L^−1^ potassium bitartrate and 200 mg L^−1^ of di-ammonium hydrogen phosphate. Aliquots of these cultures were added to the must to obtain a yeast population over 10^6^ cells mL^−1^. The fermentation processes were carried simultaneously in 2 L Pyrex glass cylinders filled with 1.75 L of the same treated must; two of them were inoculated with Y1, two with Y2 and one with WY. At the same time, 50 mg L^−1^ of bentonite were added to two of the glass cylinders inoculated with Y1 and Y2, to study its effect on the resulting wines. These experiments were carried out in triplicate and the cylinders were submerged in a thermostatic water bath at 18 °C to regulate the fermentation temperature.

Fermentation was finished when the density dropped to approx. 990 g L^−1^, which was reached after 15 days fermentation at 18 °C. Wines were analyzed and tasted after spontaneous decantation and stabilization at −2 °C for 20 days.

### 3.2. Analytical Methods

The enological variables pH, ethanol, titratable acidity, volatile acidity and reducing sugars were determined according the OIV (2021) protocols [41]. pH was measured in a Crison GLP 21+ pH-Meter. Ethanol was measured with an alcoholmeter in the distillate obtained by subjecting 200 mL of wine to steam distillation in a Selecta DE-1626 oenological distiller. The titratable acidity was obtained by titrating 10 mL of wine mixed with 10 mL of distilled water, plus a sodium hydroxide solution (0.1 N standardized), to reach pH = 7. Volatile acidity was obtained also by titration with standardized Na(OH) 0.1 N of the distillate of 20 mL of wine by steam distillation. Reducing sugars were analyzed after wine clarification with Carretz I and Carretz II reactants, followed by the addition of an alkaline copper solution, and titration with sodium thiosulfate solution (0.1 N). The absorbance at 280, 420, 520 and 620 nm was measured in an Agilent Cary 60 UV-Vis spectrophotometer (Agilent technologies, Santa Clara, CA, USA) to obtain the color intensity (CI) and Total Polyphenol Index (TPI).

Major volatile compounds and polyols were analyzed by gas chromatography in the Agilent 6890 GC (Agilent technologies, Santa Clara, CA, USA) provided with a Flame Ionization Detector (FID) and using the method of Peinado et al. 2004 [42]. Capillary column CP-WAX 57 CB (60 m long; 0,25 mm i.d.; 0,4 µm film thickness) was used. The wine sample was previously treated by adding 1 mL of a 1.018 mg L^−1^ of 4-methyl-2-pentanol (CAS number 108-11-2) solution as internal standard and 0.2 g of solid calcium carbonate to a volume of 10 mL. After, this mixture was stirred for 30 s in an ultrasonic bath and lastly subjected to centrifugation at 5000 rpm for 10 min at 2 °C temperature, to remove tartaric acid from the wine. The liquid phase was then transferred to another falcon tube and a volume of 0.7 µL was injected into the gas chromatograph inlet. Quantification of methanol, higher alcohols (1-propanol, isobutanol, 2 and 3-methyl-1-butanol and 2-phenylethanol), acetaldehyde, acetoin, ethyl acetate and the polyols glycerin and 2,3-butanodiol (*levo* and *meso* forms) was performed by using the response factors previously obtained by subjecting standard solutions of each compound to the same treatment as the samples. Pure chemicals provided by Sigma-Aldrich (St. Louis, MO, USA) and Merck (Darmstadt, Germany) were used for calibration purposes. Compound quantification was performed by means of a calibration table built with standard solutions, containing known concentrations of the target compounds. All quantified compounds in wine samples were identified and confirmed by GC-MS in an Agilent 7890 A, with a MSD-5975-C detector (Wilmington, DE, USA) using the same capillary column and settings for temperature and the carrier helium gas. Subsequently, their Linear Retention Index (LRI) was used for a second confirmation of each compound, according to previous works [33,34,43].

### 3.3. Sensorial Analysis

The wines were evaluated by a tasting panel constituted by eight trained judges (5 males and 3 females) from the Department of Agricultural Chemistry, Edaphology and Microbiology at the University of Córdoba, Spain. The panel used the tasting sheet from the OIV (2021) [41], which evaluates the attributes for sight (limpidity, aspects other than limpidity), smell (genuineness, positive intensity, quality), taste (genuineness, positive intensity, harmonious persistence, quality). Furthermore, the tasting panel was asked to give an overall quality judgement taking into account the sensory descriptors, giving a final total score ranging from 50 to 100 points. Panelists were provided with detailed instructions from the panel leader on defining these descriptors and how to proceed with sensory assessment. All samples were stored 24 h at 4 °C before the analysis. Each treatment was evaluated in a random order and the wine samples (30 mL) were presented to the tasters at room temperature (20 °C) in standardized wine glasses (NF V09-110 AFNOR, 1995), in accordance with the requirements by ISO 3591 norms.

### 3.4. Statistical Analysis

All data matrices were subjected to statistical analysis using the Statgraphics statistical software package (Centurion v. 16.1.11). Multiple Analysis of Variance (MANOVA) and Multiple Variable Analysis (MVA) were carried out to establish significant differences between the five wines obtained. A Principal Component Analysis (PCA) was carried out to provide a general interpretation of the main quantitative information contained in the data matrix of volatile compounds and polyols. Simple regression coefficients were calculated between sensory and chemical variables in order to formulate hypotheses about those potentially linked to the attributes scored in the wine sets. Lastly, Partial Least Square regression analyses (PLS) were performed to provide preliminary and easy predictive models to calculate in an objective way the scores of the wine attributes tested.

## 4. Conclusions

The statistical tests carried out on the main oenological variables show that only total acidity and reducing sugars have 5 homogeneous groups (HG) at a *p* ≤ 0.05 significance level, which agrees with the 5 wines obtained, while pH, volatile acidity, and the absorbance at 280, 420, 520 and 620 nm have 4 HGs and ethanol content, 3 HGs. The lowest value for reducing sugars was obtained in the fermentation performed with the two commercial yeasts without bentonite.

The addition of bentonite as pre-fermentative treatment significantly increases the content in higher alcohols and ethyl acetate of wines, which are also dependent on the yeast strain used.

A Multiple Variable Analysis based on the contents of ethyl acetate, methanol, 1-propanol, isobutanol, 2-methyl-1-butanol, 3-methyl-1-butanol and 2-phenylethanol provides a footprint showing the effect of bentonite on the main volatile molecules of wines. The Principal Component Analysis carried out with quantified data on all the volatile compounds and polyols resulted in two components explaining respectively 60.15% and 25.91% of the total variance and established five sample groups, in correspondence with the five wines.

The sensorial analysis shows significant differences (*p* ≤ 0.001) between the sight and smell and *p* ≤ 0.02 for the overall quality of wines obtained with the two different yeasts with the addition or not of bentonite. Only taste quality shows no significant differences. All wines were scored as good with 70–79 points, except for the wine obtained from the wild yeasts, which was scored at 66 points.

Statistically significant predictive models for taste, sight and smell based on 4, 6 and 10 chemical variables, respectively, were obtained by a Partial Least Square analysis.

This study reveals that the addition of bentonite, before the fermentation, affects in different ways the content of the main secondary metabolites and the sensory properties of wines obtained with different starter cultures of *Saccharomyces cerevisiae* yeasts.

## Figures and Tables

**Figure 1 molecules-27-08057-f001:**
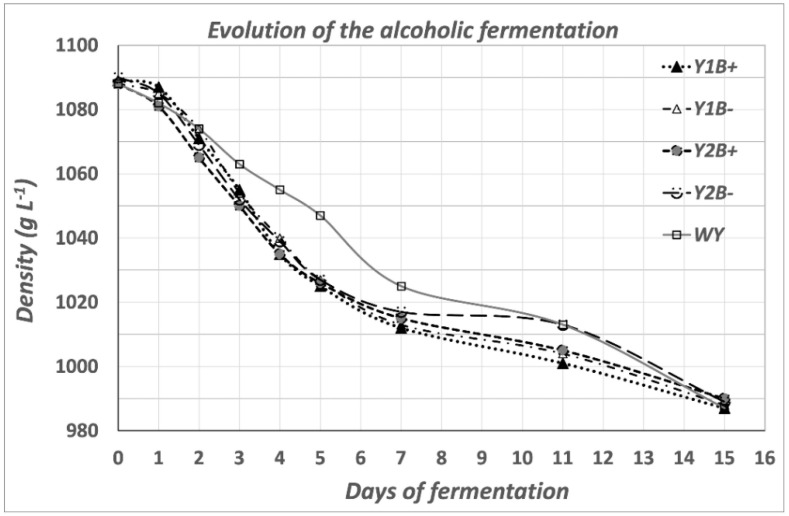
Evolution of alcoholic fermentation shown by density measurement. Yeasts tested: WY: Control (spontaneous fermentation with wild yeasts); Y1: PDM active dry yeast; Y2: Caracter active dry yeast. B+ or B− refers to the fermentation carried out with (+) or without (−) bentonite addition.

**Figure 2 molecules-27-08057-f002:**
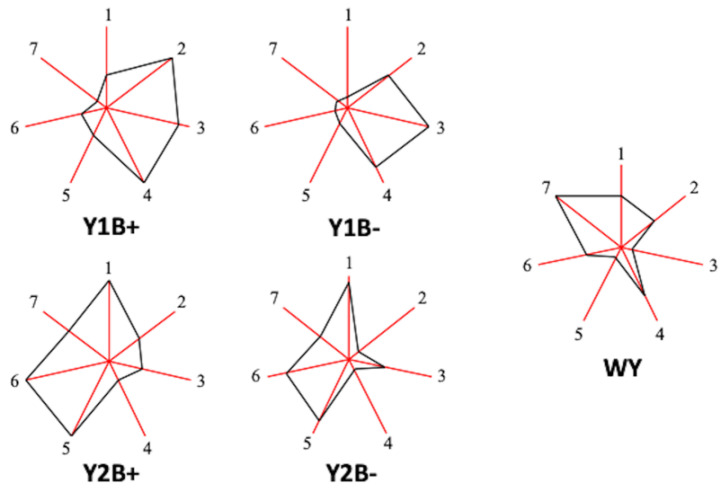
Footprints obtained from the contents of 7 volatile compounds, showing 4 or 5 homogeneous groups at the 95% confidence level. Each ray in the heptagon corresponds to one compound and the distance from the centre to each vertex to the value for each one. The end of the ray is the mean value plus three standard deviations and the origin the mean minus three standard deviations. Yeasts tested: WY: Control (spontaneous fermentation with wild yeasts); Y1: PDM active dry yeast; Y2: Caracter active dry yeast. B+ or B− refers to the fermentation carried out with (+) or without (−) bentonite addition before fermentation. Volatile compounds: 1: Ethyl acetate; 2: Methanol; 3: 1-Propanol; 4: Isobutanol; 5: 2-methyl-1-butanol; 6: 3-metyl-1-butanol; 7: 2-phenylethanol.

**Figure 3 molecules-27-08057-f003:**
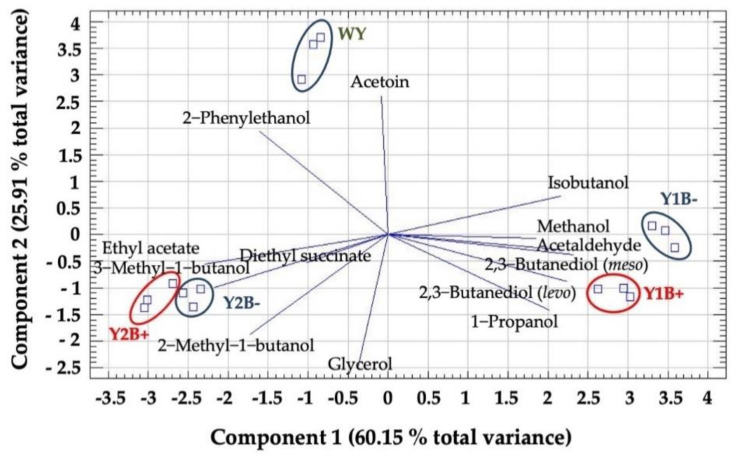
Biplot of the Principal Components Analysis carried out with the contents in major volatile compounds and polyols. Each vector corresponds to one variable and their projection over each axis corresponds to their contribution to each component. Each point in the figure corresponds to the wine sample score for each component. Yeasts tested: WY: Control (spontaneous fermentation with wild yeasts); Y1: PDM active dry yeast; Y2: Caracter active dry yeast. B+ or B− refers to the fermentation carried out with (+) or without (−) bentonite addition.

**Table 1 molecules-27-08057-t001:** Mean values and standard deviations in common oenological variables for the wines obtained.

	Y1B+	Y1B−	Y2B+	Y2B−	WY	HGs
Ethanol (% *v*/*v*)	11.8 ± 0.2 ^a^	12 ± 0.2 ^b^	12 ± 0.2 ^b^	12.5 ± 0.2 ^c^	11.9 ± 0.2 ^ab^	3
pH	3.43 ± 0.00 ^b^	3.49 ± 0.01 ^d^	3.22 ± 0.00 ^a^	3.22 ± 0.01 ^a^	3.46 ± 0.00 ^c^	4
Volatile acidity (g L^−1^)	0.34 ± 0.00 ^c^	0.40 ± 0.00 ^d^	0.24 ± 0.00 ^a^	0.24 ± 0.00 ^a^	0.26 ± 0.00 ^b^	4
Total acidity (g L^−1^)	3.97 ± 0.00 ^b^	3.81 ± 0.00 ^a^	6.94 ± 0.00 ^e^	6.60 ± 0.04 ^d^	4.04 ± 0.00 ^c^	5
Reducing sugars (g L^−1^)	1.92 ± 0.00 ^b^	2.28 ± 0.12 ^c^	1.68 ± 0.00 ^a^	2.64 ± 0.00 ^d^	3.12 ± 0.00 ^e^	5
IPT	7.26 ± 0.04 ^b^	8.48 ± 0.08 ^d^	7.41 ± 0.08 ^c^	7.53 ± 0.07 ^c^	6.93 ± 0.09 ^a^	4
Absorbance 420 nm	0.1934 ± 0.0005 ^c^	0.1656 ± 0.0004 ^b^	0.1935 ± 0.0003 ^c^	0.2170 ± 0.0009 ^d^	0.1635 ± 0.0007 ^a^	4
Absorbance 520 nm	0.0757 ± 0.0003 ^d^	0.0403 ± 0.0001 ^a^	0.0512 ± 0.0005 ^c^	0.0747 ± 0.0012 ^d^	0.0436 ± 0.0003 ^b^	4
Absorbance 620 nm	0.0483 ± 0.0005 ^d^	0.0114 ± 0.0005 ^a^	0.0154 ± 0.0006 ^b^	0.038 ± 0.002 ^c^	0.0166 ± 0.0002 ^b^	4

^a, b, c, d, e^ Different letters in the same row indicate homogeneous groups (HG) with statistical differences at 0.05 significance level. Identification of wine samples: WY: wine obtained by spontaneous fermentation; Y1: wines obtained by using starter cultures of PDM active dry yeast; Y2: wines obtained by starter cultures of Caracter active dry yeast; B+, bentonite addition to grape must; B−, no addition of bentonite.

**Table 2 molecules-27-08057-t002:** Major volatile compounds (mg L^−1^) and polyols quantified in wines obtained with different yeasts and with or without addition of bentonite.

Compounds	PubChem CID	Y1B+	Y1B−	Y2B+	Y2B−	WY	HG
Acetaldehyde	177	106 ± 8 ^c^	161 ± 4 ^d^	40 ± 2 ^a^	63 ± 1 ^b^	58 ± 5 ^b^	4
Ethyl acetate	8857	10.1 ± 0.3 ^b^	7.6 ± 0.9 ^a^	15.9 ± 0.6 ^d^	15.2 ± 0.6 ^d^	12.2 ± 0.3 ^c^	4
Methanol	137654	109 ± 8 ^d^	87 ± 3 ^c^	76 ± 2 ^b^	55 ± 2 ^a^	78 ± 3 ^b^	4
Propanol	1031	54 ± 1 ^c^	59.1 ± 0.5 ^d^	39 ± 1 ^b^	39.5 ± 0.8 ^b^	29.7 ± 0.9 ^a^	4
Isobutanol	6560	47.1 ± 0.4 ^e^	44.1 ± 0.5 ^d^	35.7 ± 0.3 ^b^	33.9 ± 0.4 ^a^	42.1 ± 0.3 ^c^	5
2-Methyl-1-butanol	8723	44 ± 1 ^c^	38.5 ± 0.7 ^b^	65 ± 3 ^e^	59.6 ± 0.3 ^d^	36.4 ± 0.4 ^a^	5
3-Methyl-1-butanol	31260	178 ± 2 ^b^	163 ± 2 ^a^	236 ± 7 ^e^	217 ± 3 ^d^	186 ± 1 ^c^	5
Acetoin	173	9 ± 2 ^a^	14 ± 1 ^c^	11 ± 1 ^bc^	9 ± 1 ^a^	20 ± 2 ^d^	4
2,3-Butanediol *levo*	225936	749 ± 56 ^b^	714 ± 29 ^b^	319 ± 36 ^a^	354 ± 25 ^a^	303 ± 72 ^a^	2
2,3-Butanediol *meso*	220010	265 ± 17 ^b^	271 ± 24 ^b^	111 ± 12 ^a^	127 ± 9 ^a^	142 ± 32 ^a^	2
Diethyl succinate	31249	21.6 ± 0.9 ^a^	21 ± 1 ^a^	23 ± 2 ^a^	22 ± 1 ^a^	21 ± 2 ^a^	1
2-Phenylethanol	7409	30 ± 3 ^a^	31.1 ± 0.7 ^a^	56 ± 3 ^c^	48 ± 5 ^b^	79 ± 5 ^d^	4
Glycerol (g L^−1^)	753	11.9 ± 0.4 ^c^	10.8 ± 0.5 ^b^	12.7 ± 0.9 ^d^	12.0 ± 0.4 ^c^	8.6 ± 0.9 ^a^	4

^a, b, c, d, e^ Different letters in the same row indicate homogeneous groups (HG) with statistical differences at 0.05 significance level. Identification of wine samples: WY: wine obtained by spontaneous fermentation; Y1: wines obtained by using starter cultures of PDM active dry yeast; Y2: wines obtained by starter cultures of Caracter active dry yeast; B+, bentonite addition to grape must; B−, no addition of bentonite.

**Table 3 molecules-27-08057-t003:** Mean scores, standard deviations, MANOVA (HGs) and ANOVA results for sensorial attributes tested.

Yeast	Y1B+		Y1B−		Y2B+		Y2B−		WY		HGs	ANOVA (*p* Values)	
Attributes	Mean	STD	Mean	STD	Mean	STD	Mean	STD	Mean	STD		Yeast	Bentonite
Sight	8.88 ^c^	0.60	8.88 ^c^	0.78	7.88 ^b^	0.60	7.38 ^ab^	0.70	6.75 ^a^	0.66	3	0.000	0.436
Smell	13.75 ^ab^	0.97	14.63 ^bc^	1.32	15.50 ^c^	1.32	15.00 ^bc^	1.58	12.38 ^a^	1.11	3	0.001	0.758
Taste	30.25 ^a^	4.18	30.38 ^a^	3.81	28.13 ^a^	3.02	28.88 ^a^	3.10	27.38 ^a^	3.94	1	0.140	0.662
Overall quality	23.00 ^b^	2.35	23.13 ^b^	3.14	21.38 ^ab^	2.12	21.75 ^ab^	2.44	19.63 ^a^	1.11	2	0.018	0.804
Total points	75.88 ^b^	6.47	77.00 ^b^	6.40	72.88 ^ab^	6.55	73.00 ^b^	6.36	66.13 ^a^	5.60	2	0.005	0.850

^a, b, c^ Different letters in the same row indicate homogeneous groups with statistical differences at 0.05 significance level. Identification of wine samples: WY: wine obtained by spontaneous fermentation; Y1: Wines obtained by using starter cultures of PDM active dry yeast; Y2: wines obtained by starter cultures of Caracter active dry yeast; B+, bentonite addition to grape must; B−, no addition of bentonite.

**Table 4 molecules-27-08057-t004:** Weights of chemical variables used to create the loading factors from each latent variable (LV) obtained for taste, sight and smell sensory attributes. Cumulative variance explained by each LV in %; *p*-value for a 95% confidence level and RMSE (Root Mean Square Error) value.

PLS for TASTE				PLS for Sight			PLS for SMELL		
	LV 1	LV 2	LV 3		LV 1	LV 2		LV 1	LV 2
Taste Qual	0.671	0.263	0.145	Sight overall	0.726	0.189	Smell Qual	0.398	0.39
% variance	45.16	84.01	99.89	% variance	17.63	65.45	% variance	38.03	72.68
*p*-value	0.00013			*p*-value	0.022		*p*-value	0.029	
RMSE	0.2			RMSE	0.133		RMSE	0.133	
**Independent Variables and Latent Variables Selected**									
**Variables**	**LV 1**	**LV 2**	**LV 3**	**Variables**	**LV 1**	**LV 2**	**Variables**	**LV 1**	**LV 2**
pH	0.158	-0.558	0.149	pH	0.115	−0.428	Acetic acid	−0.023	0.242
Acetic acid	0.531	0.029	0.629	Ethanol	−0.3	−0.414	Acetaldehyde	−0.093	0.118
Ethanol	−0.399	0.374	0.762	Abs (420)	0.101	0.507	Ethyl acetate	0.157	−0.116
Reducing sugars	−0.73	−0.74	−0.034	Abs (520)	0.113	0.43	1-Propanol	0.192	0.379
				Abs (620)	0.092	0.313	Isobutanol	0.071	0.392
				Abs (280)	0.93	0.325	3-Methyl-1-butanol	0.313	0.071
							2-Methyl-1-butanol	0.246	−0.002
							Acetoin	−0.077	0.049
							Diethyl succinate	0.871	0.771
							2-Phenylethanol	−0.034	−0.125

## Data Availability

The data presented in this study are available on request from the corresponding author.

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
