# Peer review of "Effect of Bentonite Addition to Pedro Ximénez White Grape Musts before Their Fermentation with Selected Yeasts on the Major Volatile Compounds and Polyols of Wines and Tentative Relationships with the Sensorial Evaluation"

_molecules, 2022, doi:10.3390/molecules27228057_

Round 1

Reviewer 1 Report

This work by Muñoz-Castells and collegues analyses the outcomes of bentonite addition prior to wine fermentation by means of wild or of two commercial dry yeast preparations. The chemical composition and common oenological variables are measured, as well as sensorial aspects are evaluated. Statistical analyses are performed in order to establish tentative correlations between sensory attributes and chemical composition variation.

The scope of the study is of interest and relevant for the field, however authors should clarify the idea underlying the study: why should it be better to add bentonite prior to fermentation? Please discuss this aspect.

Moreover, bentonite addition is a common practice to prevent hazes formation, and is already known to affect chemical and sensory composition of wines. Please discuss the advantages of using bentonite fining prior to fermentation.

Authors should highlight the novelty of their work compared to previous studies, such as for example “The effect of bentonite fining at different stages of white winemaking on protein stability”(2011) by Pocock and colleagues.

Additionally, according to that study in order to obtain a protein stable wine, more bentonite had to be added to the juice than when bentonite was added to fermenting must and wine. Therefore, what is the advantage envisaged by the authors in adding bentonite prior to must fermentation?

By which criteria was the amount of bentonite used in this study determined?

Identification of wine samples should be made explicit not only in table legends but also in the main text.

Analytical methods should be explained in more detail, not just citing OIV recommendations. No sufficient details were given to replicate the proposed experimental procedures and analysis.

Additionally, authors should explain why is WY considered the appropriate control.

Conclusions are consistent with the presented arguments. However the choice of adding bentonite prior to fermentation should be discussed and commented, according to the previous comments

Authors should uniform tables’ formatting (tables 1 and 2 formatting differ from tables 3 and 4)

Additionally, the use of English language and style needs to be improved.

Author Response

Reviewer 1

 We thank reviewer 1 for his/her comments and we did corrections according to them.

The manuscript has been substantially improved; we think.

Questions and answers

  1. The scope of the study is of interest and relevant for the field, however authors should clarify the idea underlying the study: why should it be better to add bentonite prior to fermentation? Please discuss this aspect.

  1. Moreover, bentonite addition is a common practice to prevent hazes formation, and is already known to affect chemical and sensory composition of wines. Please discuss the advantages of using bentonite fining prior to fermentation.

ANSWER to questions 1, 2:

According to Vela et al. (2017) clarification with bentonite to obtain wines with protein stability can be carried out in almost all stages of the winemaking process. Since the last decade of the 20th century, several authors (Ayestaran et al., 1995; Puig-Deu et al. 1999; Lambri et al., 2012) have made recommendations on the use of fining agents in white grape musts to improve the fermentation process and the wine quality and for its favorable impact on protein stabilization and the preservation of varietal aromas. Other authors, such as Pocock et al. (2011), and Lira et al. (2015) recommend the addition of bentonite during alcoholic fermentation to prevent protein haze as a clarifying and stabilizing agent because a smaller amount of product is required and consequently the removal of aroma compounds is apparently less. By contrast, Somers and Ziemelis (1973) and Puig-Deu et al. (1999) establish that bentonite treatment of finished wine is more effective in stabilizing it from the protein hazard point of view. These discrepancies show that the most appropriate moment for the addition of bentonite during the winemaking process is still not very clear, particularly to minimize its effect on wine aroma. Regarding this, Moreno and Peinado (2012) summarize the advantages of using bentonite on musts to obtain white wine as follows:

  1. It produces more compact lees because the bentonite settles out with the yeast. This means that less liquid is lost.
  2. It partly eliminates tyrosinase and therefore exerts a certain protective action against

oxidation.

  1. It stimulates fermentation as it provides a support for the yeast, which would otherwise

sink to the bottom of the tank.

  1. It adsorbs traces of fungicide that may have been transferred from the vineyard to the

must.

  1. It contributes to the production of finer, clearer wines, without interfering with aroma.

Indeed it can even enhance aroma.

  1. It minimizes the number of operations required during the winemaking process.

References:

Ayestaran BM, Ancin MC, Garcia AM, Gonzalez A and Garrido JJ. 1995. Inf luence of prefermentation clarification on nitrogenous contents of musts and wines. J Agric Food Chem 43:476-482.

Lambri M, Dordoni R, Silva A and De Faveri DM. 2012. Comparing the impact of bentonite addition for both must clarification and wine fining on the chemical profile of wine from Chambave Muscat grapes. Int J Food Sci Tech 47:1-12.

Lira E, Rodríguez-Bencomo JJ, Salazar FN, Orriols I, Fornos D and López F. 2015. Impact of bentonite additions during vinification on protein stability and volatile compounds of Albarino wines. J Agric Food Chem 63:3004-3011.

Moreno, J., Peinado, R.A.  (2012) Enological Chemistry,  pag 346-347, ISBN : 978-0-12-388438-1. Ed, Academic Press, Elsevier. (San Diego, CA. USA)

Pocock K F, Salazar FN and Waters EJ. 2011. The effect of bentonite fining at different stages of white winemaking on protein stability. Aust J Grape Wine Res 17:280-284.

Puig-Deu M, López-Tamames E, Buxaderas S and Torre-Boronat MC. 1999. Quality of base and sparkling wines as influenced by the type of fining agent added pre-fermentation. Food Chem 66:35-42.

Somers TC and Ziemelis G. 1973. The use of gel column analysis in evaluation of bentonite fining procedures. Am J Enol Vitic 24:51-54.

Vela, E., Hernández-Orte, P.,  Castro, E., Ferreira,V. and Lopez, R (2017).   Effect of Bentonite Fining on Polyfunctional Mercaptans and Other Volatile Compounds in Sauvignon blanc Wines.  Am. J. Enol. Vitic. 68 (1): 30-38.

According to your suggestions, the Introduction has been modified in the revised Manuscript.

  1. Authors should highlight the novelty of their work compared to previous studies, such as for example “The effect of bentonite fining at different stages of white winemaking on protein stability”(2011) by Pocock and colleagues.

ANSWER:

In our opinion, according to the first title of the Manuscript: “Effect of bentonite addition before fermentation with selected yeast on the major volatile compounds and polyols, tentative relations with sensorial evaluation of wines”,   this work is aimed to study the effect of bentonite in some important secondary metabolites excreted by different yeasts when bentonite remains in fermenting musts.

However, following your suggestions, we modify the title and add some sentences to the introduction. The final sentences were addressed to solve this question: 

The discrepancies described above about the appropriate moment for the addition of bentonite, together with the use of active dry yeasts, open new challenges regarding the effect of this finning agent on the fermentation process and its interaction with aroma compounds produced by yeasts.

This study aims to (1) chemically characterize and sensory evaluate the wines obtained by adding bentonite and different yeast starter cultures to the same grape must before their alcoholic fermentation, (2) study the effect of the interaction of yeast-bentonite on the major volatile compounds and polyols, and (3) relate the chemical compositions to sight, taste and smell attributes by multivariate statistical analysis”

  1. Additionally, according to that study in order to obtain a protein stable wine, more bentonite had to be added to the juice than when bentonite was added to fermenting must and wine. Therefore, what is the advantage envisaged by the authors in adding bentonite prior to must fermentation?

ANSWER:

This work was carried out on must from the white grape variety Pedro Ximénez, and it was not necessary to add more bentonite to clarify the obtained wines because they were subjected to a spontaneous stabilization at low temperature (-2 ºC) for 20 days, as is described in the material and methods section. In addition, wines from this variety are used to obtain the traditional Fino-type wines from the Southern of Spain, which show a great stability against protein haze, mainly due to the mannoproteins released by the flor-veil yeasts during the biological aging process that is conducted along several years.

For other wine types, it should make the stability assays and tests to optimize the dose of bentonite as fining agent, to reach a good stability.

Lines 111-113. This sentence has been replaced by:  “In this sense, the proposed research contributes to these advances by studying the effects on the analytical and sensory quality of wine when bentonite is added to the must and remain in it along the fermentation”.

  1. By which criteria was the amount of bentonite used in this study determined?

ANSWER

The manufacturer of bentonite recommend doses from 20 to 100 g/hL to prevent  protein haze in the stabilization of white wine, before bottling. However, several winemakers and technicians who use bentonite as a pre-fermentative agent recommend the addition of 5 g/hL to healthy white grape musts, which can be increased for rotten grapes or decreased in the case of the second fermentation of sparkling wines. This addition is aimed to diminish the content in grape proteins as well as pesticide residues coming from vine treatments and also to limit the negative effects of bentonite on the sensorial properties. Additionally, the bentonite particles can act as a physical support for the formation of yeast colonies, which favor a good development of fermentation. In our work, the bentonite added to the grape must and  kept until the natural stabilization process.

This sentence is added to the material and methods: “Bentonite suspension was prepared by adding hot water (37 ºC) to bentonite (ratio 10  to 1). The mixture was stirred for 30 min and left at room temperature for 24 hours. Then, it was added to the musts (5 g/hL) according the recommendations of experienced winemakers,  who use bentonite in Pedro Ximenez healthy grape musts fermentations”

  1. Identification of wine samples should be made explicit not only in table legends but also in the main text.

ANSWER:

An explanation of the abbreviations used for the wine samples has been added in each section of the main text, when they are cited for the first time.

  1. Analytical methods should be explained in more detail, not just citing OIV recommendations. No sufficient details were given to replicate the proposed experimental procedures and analysis.

ANSWER:

We added the following sentences in Material and Methods section to improve the description of analytical methods.

The enological variables pH, ethanol, titratable acidity, volatile acidity and reducing sugars were determined according the OIV (2021) protocols [41]. pH was measured in a Crison GLP 21+ pH-Meter. Ethanol was measured with an alcoholmeter in the distillate obtained by subjecting 200 mL of wine to steam distillation in a Selecta DE-1626 oenological distiller. The titratable acidity, was obtained by titration 10 mL of wine added with 10 mL of distilled water, with a sodium hydroxide solution 0.1 N standardized, to reach a pH=7. Volatile acidity was obtained also by titration with standardized Na(OH) 0.1 N of the distillate  of 20 mL of wine by steam distillation. Reducing sugars were analyzed after wine clarification with Carretz I and Carretz II reactants, followed by the addition of an alkaline copper  solution, and titration with sodium thiosulfate solution 0.1 N. The absorbance at 280, 420, 520 and 620 nm was measured in an Agilent Cary 60 UV-Vis spectrophotometer (Agilent technologies, Santa Clara, United States) to obtain the colour intensity (CI) and Total Polyphenol Index (TPI).

To improve the description of experiments carried out, we added the following sentence in Material and methods section:

Bentonite suspension was prepared by adding hot water (37 ºC) to bentonite (ratio 10  to 1). The mixture was stirred for 30 min and left at room temperature for 24 hours. Then, it was added to the musts (5 g/hL) according the recommendations of experienced winemakers,  who use bentonite in Pedro Ximenez healthy grape musts fermentations. This dose can be increased for rotten grapes or decreased in the case of the second fermentation of sparkling wines. Our work has consisted in adding bentonite to the must before fermentation and leaving it in the fermenting must until the spontaneous stabilization of wine at low temperature.

  1. Additionally, authors should explain why is WY considered the appropriate control.

ANSWER:

A negative control without the addition of yeasts nor bentonite was used, aimed to compare the effect of both variance sources (yeast inoculum and bentonite addition). These type of control is frequently used in winemaking experiments and works published in relevant journals, as the following papers are:  http://dx.doi.org/10.1016/j.foodchem.2017.07.163  and https://www.ajevonline.org/content/68/1/30

The following text is added in material and methods section:

“This fermentation was used as a negative control to compare the effects of both variables on volatile compounds, polyols, and sensory analysis. This type of control is widely used by winemakers in relevant studies to verify the effect of the tested variable (Dumitru et al., 2018; Vela et al., 2017)”

  1. Conclusions are consistent with the presented arguments. However the choice of adding bentonite prior to fermentation should be discussed and commented, according to the previous comments.

ANSWER

The following lines are added to the discussion section after line 693 of the revised manuscript:

“Several studies show that the effect of bentonite addition to white grape must is variable and dependent of grape variety. Regarding this, Armada and Falque, (2007) [30] found no significant differences but Lambri et al. (2010) [3] reported clear decreases in ethyl butyrate or hexanoate associated with bentonite. Even higher quantities of some compounds in musts fined with bentonites are found in other studies[10,11,36]. Vela et al. (2017) [5] found in Sauvignon blanc that only butyl acetate, ethyl hydrocinnamate and ethyl cinnamate showed significant differences linked to bentonite treatment, while other ethyl esters and acetates did not show significant differences. These results agree with those reported previously [3,16], suggesting that the vintage and bentonite type are major sources of variability.

In addition, the results obtained in the present work show that bentonite affect in different ways according to the inoculum format used for the fermentation of Pedro Ximénez grape musts, providing different foot-printings at a significant level (p-value= 0,05) for the content in methanol, 5 higher alcohols and ethyl acetate in wines.

Furthermore, the use of advanced statistical tools such as Principal Component analysis, applied to all volatile compounds and quantified polyols, allows to establish differences between wines treated with bentonite for each inoculum tested, based on the scores of each wine two components. Through this analysis, a smaller difference is obtained between the wines from PDM yeast than between wines from character yeast with or without bentonite”

  1. Authors should uniform tables’ formatting (tables 1 and 2 formatting differ from tables 3 and 4)

ANSWER:

The format of tables has been corrected.

  1. Additionally, the use of English language and style needs to be improved.

ANSWER:

The English language has been carefully revised and improved by an English translator specialist in wine microbiology research.

Reviewer 2 Report

I am carefully reviewing the manuscript “Effect of Bentonite Addition before Must Fermentation with Selected Yeasts on the Major Volatile Compounds, Polyols and Tentative Relations with Sensorial Evaluation of Wines”. In this manuscript,  the addition of bentonite

In pre-fermentation period was well studied. Major volatile compounds, polyols and sensory data were observed and discussed. This work is interesting and important for winemaking and additives use. From my opinion, this manuscript need revision before published.

-Title

As far as I learned from your manuscript, your experiment was conducted in ‘white wine’. It is important for the bentonite addition treatment before fermentation. It’s better to clarify the variety or mark it as white wines in title.

-Abstract and introduction

why did you use bentonite before fermentation rather than after? This is important and the most interesting point in your manuscript. But I donnot think that it was clearly presented in abstract and introduction.

-Results and discussion

All the figures and tables were not in well form. Please re-organize them.

-Methods

Chemicals should be well described in the text. Especially for the GC standards.

P425-433

Sensory analysis was ambiguous. Which attributors were selected? How do the terms generate? Did the sensory experiment reproduce……

Author Response

Reviewer 2

We thank reviewer 2  for his/her comments and we did corrections according to them.

The manuscript has been substantially improved; we think.

Questions and answers

 Comments and Suggestions for Authors

I am carefully reviewing the manuscript “Effect of Bentonite Addition before Must Fermentation with Selected Yeasts on the Major Volatile Compounds, Polyols and Tentative Relations with Sensorial Evaluation of Wines”. In this manuscript,  the addition of bentonite in pre-fermentation period was well studied. Major volatile compounds, polyols and sensory data were observed and discussed. This work is interesting and important for winemaking and additives use. From my opinion, this manuscript need revision before published.

  1. -Title

As far as I learned from your manuscript, your experiment was conducted in ‘white wine’. It is important for the bentonite addition treatment before fermentation. It’s better to clarify the variety or mark it as white wines in title.

Answer:

Title has been modified as follows: “Effect of bentonite addition to Pedro Ximénez white grape musts before their fermentation with selected yeasts on the major volatile compounds and polyols of wines and tentative relationships with the sensorial evaluation”  

See changes in Revised Manuscript

  1. -Abstract and introduction

why did you use bentonite before fermentation rather than after? This is important and the most interesting point in your manuscript. But I donnot think that it was clearly presented in abstract and introduction.

Answer:

Abstract and Introduction are modified to improve their understanding.

Abstract section:

 In this work, we study the effect of bentonite addition to the grape must before alcoholic fermentation on the chemical composition and sensorial profile of the obtained wines. Fermentations were carried out with two Saccharomyces cerevisiae commercial active dry yeasts treated or not with bentonite and were compared with a control wine obtained by spontaneous fermentation (using the grape must microbiota).

This section has been substantially improved taking into account the suggestions of both reviewers.

  1. -Results and discussion

All the figures and tables were not in well form. Please re-organize them.

Answer:

The format of tables and figures has been corrected.

  1. -Methods

Chemicals should be well described in the text. Especially for the GC standards.

 Answer:

The chemical products suppliers have been added in the material and methods section.

Each volatile compound was identified in Table 2 by the PubChem CID number, which is the identification number in the database of chemical molecules and their activities in biological tests (Compound ID number).

The following sentence is added in Material and Methods section:

Pure chemicals provided by Sigma-Aldrich (St. Louis, Mo., USA) and Merck (Darmstadt, Germany) were used for calibration purposes. Compound quantification was performed by means of a calibration table built with standards solutions, containing known concentrations of the target compounds.

  1. Sensory analysis was ambiguous. Which attributors were selected? How do the terms generate? Did the sensory experiment reproduce……

Answer:

Attributes were selected according to the sheet provided in the publication of International Organisation  of Wine and Vine. (OIV) in 2021: (https://www.oiv.int/public/medias/7895/oiv-patronage-competition-norme-ed-2021.pdf)  OIV standard for international wine and spirituous beverages of vitivinicultural origin competitions.

The Sensorial analysis section was corrected as follows:

“The wines were evaluated by a tasting panel constituted by eight trained judges (5 males and 3 females) from the Department of Agricultural Chemistry, Edaphology and Microbiology at the University of Córdoba, Spain. The panel used the tasting sheet from the OIV (2021) [41], which evaluates the attributes for sight (limpidity, aspect other than limpidity), smell (genuineness, positive intensity, quality), taste (genuineness, positive intensity, harmonious persistence, quality). Furthermore, the tasting panel was asked to give an overall quality judgement taking into account the sensory descriptors, giving a final total score ranging from 50 to 100 points. Panelists were provided with detailed instructions from the panel leader on defining these descriptors and how to proceed with sensory assessment. All samples were stored 24 hours at 4 ºC before the analysis. Each treatment was evaluated in a random order and the wines samples (30 mL) were presented at room temperature (20 ºC) to the tasters in standardized wine glasses (NF V09-110 AFNOR, 1995) in accordance with the requirements by ISO 3591 normative”

In the discussion section, this sentence was also added:

Line 351. … by the sum of attributes evaluated for sight, smell and taste (overall quality)…

Round 2

Reviewer 2 Report

Accept in present form